# Death of an offspring and parental risk of ischemic heart diseases: A population-based cohort study

Dang Wei[1]*, Imre Janszky[1,2], Fang Fang[3], Hua Chen[1], Rickard Ljung[3,4], Jiangwei Sun[3], Jiong Li[5], Krisztina D. László[1]

1 Department of Global Public Health, Karolinska Institutet, Stockholm, Sweden, 2 Department of Public Health and Nursing, Norwegian University of Science and Technology, Trondheim, Norway, 3 Institute of Environmental Medicine, Karolinska Institutet, Stockholm, Sweden, 4 Swedish Medical Products Agency, Uppsala, Sweden, 5 Department of Clinical Medicine - Department of Clinical Epidemiology, Aarhus University, Aarhus, Denmark

* dang.wei@ki.se

## Abstract

### Background

The death of a child is an extreme life event with potentially long-term health consequences. Knowledge about its association with ischemic heart diseases (IHDs) and acute myocardial infarction (AMI), however, is very limited. We investigated whether the death of an offspring is associated with the risk of IHD and AMI.

### Methods and findings

We studied parents of live-born children recorded in the Danish (1973 to 2016) and the Swedish (1973 to 2014) Medical Birth Registers ($n = 6,711,952$; mean age at baseline 31 years, 53% women). We retrieved information on exposure, outcomes, and covariates by linking individual-level information from several nationwide registers. We analyzed the abovementioned associations using Poisson regression. A total of 126,522 (1.9%) parents lost at least 1 child during the study period. Bereaved parents had a higher risk of IHD and AMI than the nonbereaved [incidence rate ratios (IRRs) (95% confidence intervals (CIs)): 1.20 (1.18 to 1.23), $P < 0.001$ and 1.21 (1.17 to 1.25), $P < 0.001$, respectively]. The association was present not only in case of losses due to CVD or other natural causes, but also in case of unnatural deaths. The AMI risk was highest in the first week after the loss [IRR (95% CI): 3.67 (2.08 to 6.46), $P < 0.001$], but a 20% to 40% increased risk was observed throughout the whole follow-up period. Study limitations include the possibility of residual confounding by socioeconomic, lifestyle, or health-related factors and the potentially limited generalizability of our findings outside Scandinavia.

### Conclusions

The death of an offspring was associated with an increased risk of IHD and AMI. The finding that the association was present also in case of losses due to unnatural causes, which are

**Data Availability Statement:** All the data used in the present study were obtained from the Swedish National Board of Health and Welfare (https://www.socialstyrelsen.se/en/about-us/contact-us/),

Statistics Sweden (https://www.scb.se), and Statistics Denmark (https://www.dst.dk/en/kontakt). The data cannot be shared publicly due to the Danish and Swedish relevant laws and regulations and due to ethical considerations. The data may be requested for research purposes from the above data holder authorities by researchers who fulfill specific requirements.

**Funding:** KDL was supported by the Swedish Council for Working Life and Social Research (grant no. 2015-00837), the Karolinska Institutet's Research Foundation (grants no. 2018-01924) and the Swedish Heart and Lung Foundation (grant no. 20180306). DW was supported by the China Scholarship Council (grant no. 201700260276). IJ was supported by the Karolinska Institutet's Research Foundation (grants no. 2018-01547 and 2020-01600). JL was supported by the Novo Nordisk Foundation (grant no. NNF18OC0052029), the Nordic Cancer Union (grant no. R275-A15770), the Danish Council for Independent Research (grants no. DFF-6110-00019B and 9039-00010B), and the Karen Elise Jensens Fond (grant no. 2016). FF was supported by Senior Researcher Award at Karolinska Institutet and Strategic Research Area in Epidemiology at Karolinska Institutet. HC was supported by the China Scholarship Council (grant no. 201700260296). JS was supported by the China Scholarship Council (grant no. 201700260278). The funders had no role in study design, data collection, data cleaning, data analysis, data interpretation, or writing of the report.

**Competing interests:** I have read the journal's policy and the authors of this manuscript have the following competing interests: KDL received research grants from the Swedish Council for Working Life and Social Research, the Swedish Heart and Lung Foundation, the Karolinska Institutet's Research Foundation, the Clas Groschinsky Memorial Foundation and the Swedish Society of Medicine during the past five years. RL is employed at the Swedish Medical Products Agency, Uppsala, Sweden. The views expressed in this paper do not necessarily represent the views of the Government agency. The remaining authors have nothing to disclose.

**Abbreviations:** AMI, acute myocardial infarction; CI, confidence interval; CVD, cardiovascular disease; ICD, International Classification of Diseases; IHD, ischemic heart disease; IR, incidence rate; IRR, incidence rate ratio; MBR, Medical Birth Register; SD, standard deviation; STROBE, Strengthening the Reporting of Observational Studies in Epidemiology guidelines.

less likely to be confounded by cardiovascular risk factors clustering in families, suggests that stress-related mechanisms may also contribute to the observed associations.

## Author summary

### Why was this study done?

- The death of a child is an extremely stressful life event, with potential consequences for parents' cardiovascular health. Knowledge about the association between the loss of an offspring and the risk of ischemic heart diseases (IHDs) and acute myocardial infarction (AMI), however, is very limited.

- If our hypotheses regarding such associations are correct, this knowledge may raise awareness about the importance of support in bereavement and of increased cardiovascular risk monitoring for bereaved parents.

### What did the researchers do and find?

- We conducted a population-based cohort study including 6.7 million parents from Denmark and Sweden to investigate the association between loss of an offspring and the parents' risk of IHD and AMI.

- We found that parents who lost a child had higher risks of IHD and AMI than their unexposed counterparts. The associations were present not only if the offspring died due to cardiovascular and other natural causes, but also in case of unnatural deaths.

- The risk of AMI among bereaved parents was more than 3 times higher in the week after the death of a child and around 20% higher on the long term compared to that among nonbereaved parents.

### What do these findings mean?

- The findings that the association was present also in case of losses due to unnatural causes, which are less likely to be confounded by cardiovascular risk factors clustering in families, and that the risk of AMI was highest during the week after the loss suggest that stress-related mechanisms may contribute to the observed associations.

- If confirmed in other studies, our findings may call for support from family, friends, and health professionals and for increased surveillance for IHD for bereaved parents, in particular in the first week after the loss.

## Introduction

An increasing number of studies have suggested that the death of a spouse in middle and old age may increase the risk of cardiovascular mortality [1–4], particularly in the months following the loss [1,3]. Knowledge about the effect of bereavement on incident cardiovascular

diseases (CVDs) and of other types of losses is much more limited, though recently, a few studies have reported associations between the death of a spouse, a sibling, a child, or "a significant person" and increased risks of acute myocardial infarction (AMI), stroke, atrial fibrillation, and/or cardiovascular mortality [5–11].

As one of the most extreme forms of bereavement [12], the death of a child is considered a "catastrophic stressor" and is rated 6 on a 6-step scale by a widely used classification system of sources of stress [13]. Accepting the loss of a child—a task that according to several theories of bereavement is the most important for grief resolution—is very difficult for parents [14], as child mortality is very low in Western societies and is in sharp contrast with expectations about the life cycle. Parental grief is unusually intense and persistent and can hardly be fully resolved [11,15]. The loss of a child is more likely to result in complicated grief than the loss of other family members [16]. Compared to unexposed parents, bereaved parents have higher rates of mental illness [17], stress-related somatic diseases [18–21], and mortality [22–24] for several years after the loss.

To our knowledge, the association between the death of a child and the risk of incident AMI has been investigated only in one study [7]. Li and colleagues found an increased risk of AMI [7] among parents who lost a child younger than 18 years; the increased risk appeared only from the seventh year of follow-up [7]. Given the low prevalence of exposure, the low incidence of atherosclerotic CVDs in early and mid-adulthood, and the observed modest association, the study of Li and colleagues did not have sufficient statistical power to analyze the importance of the type of death, potential effect modifiers of this association, nor short-term effects after the loss; a few earlier studies have suggested that the death of other type of family members may trigger AMI [5,8]. Furthermore, the death of an adult child—not considered in the study of Li and colleagues [7]—could also be associated with an increased risk of adverse health outcomes [18,23,24]. In addition to AMI, the death of a child may increase the risk of other, less severe ischemic heart diseases (IHDs) such as angina pectoris and atherosclerosis that may eventually lead to an AMI. Modern medical technology allows an early detection of IHD, with implications for the prevention of AMI. However, the association between the death of a child and the risk of overall IHDs has not yet been studied.

In this population-based study using nationwide data from Denmark and Sweden, we investigated whether the death of an offspring is associated with an increased risk of IHD and AMI and whether these associations differ according to the time since loss, characteristics of the loss, and parental sociodemographic factors.

## Methods

### Study population and study design

We studied parents of live-born children included in the Danish Medical Birth Register (MBR) during 1973 to 2016 (*n* = 2,807,548) and in the Swedish MBR during 1973 to 2014 (*n* = 3,924,237) [11]. Children registered in the MBRs were linked to their parents by means of the Civil Registration System in Denmark and the Multi-Generation Register in Sweden. Information on mothers was available for virtually all children; we had information on fathers for almost all the Danish children and for 83% of the Swedish children, resulting in 6,731,785 parents being included in the parental cohort. We also identified children of these parents that were born before 1973, or outside Denmark/Sweden, but who were registered later in these countries.

To obtain information on parents and their family members, we linked the cohort to several other registers (S1 Table) using the unique personal identification number [11]. Since the coverage of the Danish National Hospital Register became complete in 1978 and that of the

Swedish Patient Register in 1987, we defined the study period as 1978 to 2016 for the Danish and 1987 to 2014 for the Swedish parents [11]. Parents entered the cohort at the start of the study period (January 1, 1978 in Denmark and January 1, 1987 in Sweden) if they had at least 1 child at that time, otherwise on the date of birth of the first child in Denmark/Sweden or upon immigration with child(ren) to these countries, whichever came later during the study period (S1 Fig). Parents were included in our analyses if, at the start of the follow-up, they (1) were alive and resided in Denmark or Sweden; (2) had at least 1 live child; and (3) had no record of IHD [11]. Follow-up ended at the time of the first diagnosis of IHD/AMI, death, emigration, or December 31, 2016 (Denmark) or December 31, 2014 (Sweden), whichever came first.

The study was approved by the Danish Data Protection Agency in Copenhagen and the Ethics Review Board in Stockholm. Our prespecified analysis plan is presented in the Supporting information (S1 Analysis plan). We followed the Strengthening the Reporting of Observational Studies in Epidemiology guidelines when writing the manuscript (S1 STROBE Checklist).

## Exposure

We obtained information on children's death from the Civil Registration System in Denmark and from the Cause of Death Register in Sweden. We defined exposure as death of a child after cohort entry and treated it as a time-varying variable, i.e., parents who lost a child contributed person-time to the unexposed group until the child's death and to the exposed group afterwards [11]. Parents who did not lose a child contributed person-time only to the unexposed group. In case of losses of several children during the follow-up, we considered the first loss. We classified bereaved parents according to the child's main cause of death (due to CVD, other natural causes, or unnatural causes, using the International Classification of Diseases (ICD) codes in S2 Table), the children's age at loss (≤1, 2 to 12, 13 to 18, 19 to 29, or >29 years) and the number of live children at the time of loss (0, 1 to 2, or ≥3) [11].

## Outcomes

The outcomes were a main diagnosis or death due to IHD or AMI as retrieved from the National Hospital Register and the Civil Registration System in Denmark and from the Patient Register and the Cause of Death Register in Sweden using the ICD codes presented in S2 Table.

## Covariates

We obtained information on study participants' demographic characteristics, including sex, age, country of birth, education, marital status, and income from several nationwide registries [11], as described in S1 Table. We classified income based on the tertile distribution of each 10-year interval. We defined marital status, highest education, and income based on information from the year before cohort entry. If information for the corresponding year was lacking, we used data from the year closest to study entry by checking back up to the fifth year [11]. For the Danish participants who entered the study in 1980 or earlier, we used information on income from 1980 (the first year with available data). For the Swedish participants who entered the cohort in 1990 or earlier, we used information on education from 1990 (the first year with available data) [11].

We defined study participants' history of CVD (except for IHD) and psychiatric disorders and on family (i.e., parents' and siblings') history of CVD at baseline using information from several nationwide registries on healthcare and death (S1 Table). For the Swedish mothers, we

retrieved information on pregestational or gestational hypertension and diabetes and on smoking and obesity in early pregnancy from the Swedish MBR [11]. For the Danish mothers, we obtained information on diagnoses of pregestational or gestational hypertension and diabetes from the Danish National Hospital Register and on smoking and obesity in early pregnancy from the Danish MBR. The ICD codes used to identify these medical conditions are presented in S2 Table.

## Statistical analyses

We compared baseline characteristics of the exposed and the unexposed groups by means of Student *t* tests in case of continuous variables and chi-squared tests in case of categorical variables. We used Poisson regression to estimate incidence rate ratios (IRRs) for the association between the death of a child and parental risk of incident IHD and AMI. Although we included almost all parents who gave birth to at least a child in Denmark or Sweden during our study period, we also calculated 95% confidence intervals (CIs) to infer our findings to a theoretical larger group of potential parents.

We performed analyses with any loss and with exposure categorized according to the deceased child's (1) cause of death; (2) age at death; and (3) the number of other children the parent had at the time of loss [11]. We performed 3 type of models: model 1, adjusted for age at follow-up (split at 5 years), model 2, adjusted for age (split at 5 years) and calendar year at follow-up (split at 10 years) as time-varying variables and sex, country of birth, highest education, and history of psychiatric disorders and of CVD as time-fixed variables, while model 3, adjusted for marital status, income, parents' history of CVD, and siblings' history of CVD (i.e., confounders with a high rate of missing data) in addition to variables in model 2. Criteria for including confounders in our multivariable models were (1) a known or an a priori considered plausible association with the death of a child and the risk of IHD and AMI; and (2) not being on the pathway between exposure and the outcome [25].

To visualize the changing pattern of the risk of AMI after bereavement, we performed analyses according to time since the loss (≤7 days, 8 to 30 days, 1 to 3 months, 3 to 12 months, 1 to 5 years, 5 to 10 years, or ≥10 years) [11]. In addition, we conducted a self-matched case-crossover analysis to test the hypothesis regarding a triggering effect of child's death on AMI. The hazard period was defined as 0 to 1, 2 to 7, 0 to 7, or 0 to 30 days before AMI. The control period was defined using 2 approaches: (1) the usual frequency of a child's death was calculated based on the period 30 to 180 or 30 to 365 days before the AMI; (2) the same days of the week, week of month, or month of year prior to AMI corresponding to the hazard period (S2 Fig). We used conditional logistic regression models to estimate relative risks and 95% CIs for the association of interest. Since each patient's exposure period is matched to his/her own control period, the design may eliminate confounding by risk factors that were constant within study participants during the exposure and control periods.

To investigate effect modification by country, sex, age, and education, we ran analyses stratified by these variables. To explore the robustness of our results, we performed several sensitivity analyses, i.e., we (1) excluded study participants who lost a child before baseline; and (2) adjusted for smoking and obesity in early pregnancy and pregestational and gestational hypertension and diabetes before baseline among women with data on these variables [11]. To address concerns about missing data for some covariates, we imputed missing data by 5 replications through the fully conditional specification using logistic regression [26]. After the multiple imputation for the missing data, we reran models 2 and 3.

We performed analyses using SAS 9.4 (SAS Institute Inc., Cary, NC, US).

## Results

A total of 126,522 parents (1.9% of the study participants) lost at least 1 child during the study period (Fig 1). A comparison between the exposed and unexposed parents is presented in Table 1.

During the median 21-year follow-up, 297,399 parents were diagnosed with IHD and 146,739 with an AMI. Bereaved parents had higher risks of IHD and AMI than the nonbereaved parents; the adjusted IRRs and 95% CIs were [1.20 (1.18 to 1.23), $P < 0.001$] and [1.21 (1.17 to 1.25), $P < 0.001$], respectively. The risks of IHD and AMI were highest if the child died due to CVD but were increased also after losses due to other natural or unnatural causes (Table 2). There was a trend towards a U-shaped association between the deceased child's age at loss and the parent's risk of IHD and AMI. The associations were slightly stronger among parents with 3 or more children at the time of loss than the parents with fewer children (Table 2). Bereaved

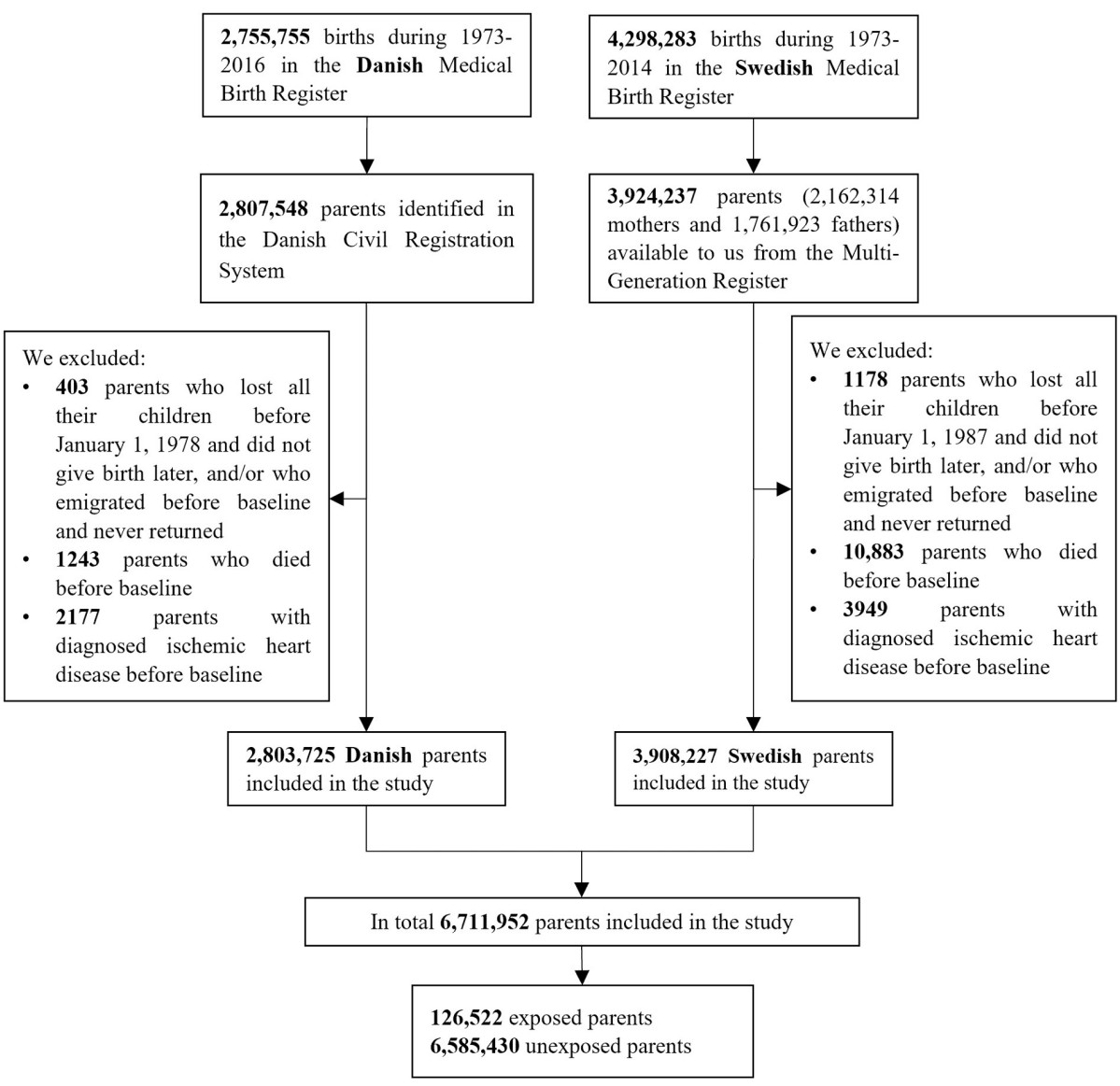

**Fig 1. Flow chart of the study.**

**Table 1. Characteristics of the study population according to the death of a child.**

| Variables | Exposed to the death of a child | | |
| --- | --- | --- | --- |
| | Unexposed ($n$ = 6,585,430) | Exposed ($n$ = 126,522) | $P^*$ |
| | N (%) | N (%) | |
| Age/Mean (SD), years | 30.7 (6.6) | 31.2 (7.7) | <0.001 |
| Sex | | | <0.001 |
| Men | 3,085,769 (46.9) | 56,743 (44.8) | |
| Women | 3,499,661 (53.1) | 69,779 (55.2) | |
| Country of birth | | | <0.001 |
| Denmark or Sweden | 5,770,503 (87.6) | 112,364 (88.8) | |
| Other countries | 814,927 (12.4) | 14,158 (11.2) | |
| Year of entry in the study | | | <0.001 |
| Before 1980 | 699,782 (10.6) | 30,495 (24.1) | |
| 1980–1989 | 2,302,881 (35.0) | 62,555 (49.4) | |
| 1990–1999 | 1,369,476 (20.8) | 19,892 (15.7) | |
| 2000–2009 | 1,381,774 (21.0) | 10,406 (8.2) | |
| After 2009 | 831,517 (12.6) | 3,174 (2.5) | |
| Marital status at baseline | | | <0.001 |
| Married or in registered partnership | 3,002,351 (45.6) | 65,738 (52.0) | |
| Single, widowed, or divorced | 2,577,992 (39.1) | 37,437 (29.6) | |
| Missing | 1,005,087 (15.3) | 23,347 (18.4) | |
| Highest education at baseline | | | <0.001 |
| 0–9 years | 1,416,750 (21.5) | 44,124 (34.9) | |
| 10–14 years | 3,584,453 (54.4) | 61,633 (48.7) | |
| ≥15 years | 1,266,295 (19.2) | 15,374 (12.2) | |
| Missing | 317,932 (4.8) | 5,391 (4.3) | |
| Income | | | <0.001 |
| Low tertile | 1,872,611 (28.4) | 34,204 (27.0) | |
| Middle tertile | 1,877,245 (28.5) | 29,699 (23.5) | |
| High tertile | 1,881,875 (28.6) | 27,301 (21.6) | |
| Missing | 953,699 (14.5) | 35,348 (27.9) | |
| History of CVD at baseline[†] | | | <0.001 |
| No | 6,435,984 (97.7) | 124,072 (98.1) | |
| Yes | 149,446 (2.3) | 2,450 (1.9) | |
| History of psychiatric disorders at baseline | | | <0.001 |
| No | 6,322,878 (96.0) | 121,903 (96.3) | |
| Yes | 262,552 (4.00) | 4,619 (3.7) | |
| Parents' history of CVD | | | <0.001 |
| No | 3,787,823 (57.5) | 62,495 (49.4) | |
| Yes | 1,580,759 (24.0) | 27,903 (22.1) | |
| Missing | 1,216,848 (18.5) | 36,189 (28.6) | |
| Sibling's history of CVD | | | <0.001 |
| No | 5,213,542 (79.2) | 88,213 (69.7) | |
| Yes | 155,040 (2.4) | 2,185 (1.7) | |
| Missing | 1,216,848 (18.5) | 36,189 (28.6) | |
| Hypertension before or during pregnancy at baseline[‡] | | | <0.001 |
| No | 3,371,167 (96.3) | 67,452 (96.7) | |
| Yes | 114,447 (3.3) | 1,835 (2.6) | |
| Missing | 14,047 (0.4) | 492 (0.7) | |

*(Continued)*

**Table 1.** (Continued)

| Variables | Exposed to the death of a child | | P* |
|---|---|---|---|
| | Unexposed (n = 6,585,430) | Exposed (n = 126,522) | |
| | N (%) | N (%) | |
| Diabetes before or during pregnancy at baseline‡ | | | <0.001 |
| No | 3,460,882 (98.9) | 68,911 (98.8) | |
| Yes | 24,732 (0.7) | 376 (0.5) | |
| Missing | 14,047 (0.4) | 492 (0.7) | |
| Smoking in early pregnancy at baseline‡ | | | <0.001 |
| No | 1,845,976 (52.7) | 19,519 (28.0) | |
| Yes | 397,433 (11.4) | 8,105 (11.6) | |
| Missing | 1,256,252 (35.9) | 42,155 (60.4) | |
| Obesity in early pregnancy at baseline‡ | | | 0.006 |
| No | 1,515,034 (43.3) | 16,615 (23.8) | |
| Yes | 125,263 (3.6) | 1,267 (1.8) | |
| Missing | 1,859,364 (53.1) | 51,897 (74.4) | |

CVD, cardiovascular diseases; IHD, ischemic heart disease; SD, standard deviation.

*The p-values corresponds to differences between the exposure groups in Student t tests in case of continuous variables and chi-squared tests in case of categorical variables.

†Except for IHDs.

‡Available only for the women.

parents had a more than 3-fold increased risk of AMI in the first week after the death of a child [IRR (95% CI): 3.67 (2.08 to 6.46), $P < 0.001$] and about 20% to 40% increased risk during the rest of the follow-up (except during 8 to 30 days after the loss) relative to nonbereaved (Fig 2). Similarly, in the case-crossover analysis, we observed 2.4 to 3 times higher AMI risk in the first week after the death of a child than in the control periods (S3 Table).

The associations between the death of a child and the risk of IHD and AMI were somewhat stronger among mothers than fathers, in Sweden than in Denmark, and among parents aged <50 years than those older (Table 3); there was no evidence of effect modification by education. The associations between the death of a child and the risk of IHD and AMI did not change substantially after (1) excluding study participants who lost a child before baseline, (2) controlling for pregestational and gestational hypertension and diabetes and maternal obesity in early pregnancy among women; the associations were slightly attenuated after adjusting for maternal smoking in early pregnancy, or (3) multiple imputation for missing data (S4 Table).

## Discussion

We found that the death of a child was associated with an increased risk of IHD and AMI. The associations were present not only when the loss was due to CVD or other natural causes, but also in case of unnatural deaths. The risks of IHD and AMI were slightly higher after loss of an infant or an adult child and if the parent had 3 or more remaining live children, compared with other losses. The risk of AMI was highest in the first week after the loss, but a modestly increased AMI risk persisted throughout the follow-up.

### Comparison with earlier studies

Our finding that the death of a child was associated with an increased risk of IHD and AMI is consistent with earlier results that death of a spouse [5], parent [27], sibling [6], or a significant

**Table 2. Adjusted IRRs and 95% CIs for IHD according to the death of a child.**

| Exposure | IHD | | | | | | AMI | | | | | |
|---|---|---|---|---|---|---|---|---|---|---|---|---|
| | Rate* | Model 1† (N = 6,711,952) | | Model 2‡ (N = 6,388,629) | | Model 3‖ (N = 4,512,391) | | Rate* | Model 1† (N = 6,711,952) | | Model 2‡ (N = 6,388,629) | | Model 3‖ (N = 4,512,391) |
| | | IRR (95% CI) | P | IRR (95% CI) | P | IRR (95% CI) | P | | IRR (95% CI) | P | IRR (95% CI) | P | IRR (95% CI) | P |
| **Unexposed** | 222.5 | 1.00 | | 1.00 | | 1.00 | | 108.7 | 1.00 | | 1.00 | | 1.00 | |
| **All deaths** | 418.4 | 1.21 (1.18–1.24) | <0.001 | 1.20 (1.18–1.23) | <0.001 | 1.21 (1.17–1.26) | <0.001 | 205.7 | 1.18 (1.15–1.22) | <0.001 | 1.21 (1.17–1.25) | <0.001 | 1.22 (1.16–1.28) | <0.001 |
| **Cause of the child's death** | | | | | | | | | | | | | |
| Death due to CVD | 688.3 | 1.29 (1.17–1.42) | <0.001 | 1.32 (1.20–1.46) | <0.001 | 1.28 (1.11–1.49) | <0.001 | 381.8 | 1.39 (1.22–1.58) | <0.001 | 1.45 (1.27–1.66) | <0.001 | 1.40 (1.16–1.68) | <0.001 |
| Other natural death | 375.8 | 1.22 (1.18–1.25) | <0.001 | 1.21 (1.18–1.25) | <0.001 | 1.20 (1.14–1.26) | <0.001 | 180.5 | 1.17 (1.12–1.21) | <0.001 | 1.18 (1.14–1.23) | <0.001 | 1.21 (1.13–1.30) | <0.001 |
| Unnatural death | 553.8 | 1.17 (1.12–1.23) | <0.001 | 1.17 (1.12–1.23) | <0.001 | 1.21 (1.15–1.28) | <0.001 | 281.3 | 1.19 (1.12–1.27) | <0.001 | 1.23 (1.15–1.31) | <0.001 | 1.21 (1.12–1.30) | <0.001 |
| **Age of the deceased child at loss** | | | | | | | | | | | | | |
| ≤1 | 224.4 | 1.31 (1.25–1.37) | <0.001 | 1.28 (1.22–1.34) | <0.001 | 1.29 (1.20–1.40) | <0.001 | 101.3 | 1.23 (1.15–1.32) | <0.001 | 1.22 (1.14–1.31) | <0.001 | 1.26 (1.13–1.41) | <0.001 |
| 2–12 | 291.2 | 1.15 (1.07–1.23) | <0.001 | 1.12 (1.05–1.20) | <0.001 | 1.10 (0.98–1.24) | 0.115 | 133.3 | 1.08 (0.98–1.19) | 0.118 | 1.06 (0.97–1.17) | 0.213 | 1.14 (0.96–1.34) | 0.131 |
| 13–18 | 460.8 | 1.12 (1.04–1.21) | 0.003 | 1.11 (1.03–1.20) | 0.004 | 1.07 (0.96–1.21) | 0.228 | 213.2 | 1.06 (0.95–1.18) | 0.324 | 1.06 (0.95–1.18) | 0.285 | 1.07 (0.91–1.25) | 0.432 |
| 19–29 | 643.3 | 1.18 (1.14–1.23) | <0.001 | 1.18 (1.13–1.23) | <0.001 | 1.20 (1.13–1.27) | <0.001 | 318.6 | 1.18 (1.12–1.25) | <0.001 | 1.20 (1.13–1.27) | <0.001 | 1.17 (1.08–1.27) | <0.001 |
| >29 | 1035.1 | 1.22 (1.17–1.28) | <0.001 | 1.27 (1.21–1.34) | <0.001 | 1.27 (1.18–1.37) | <0.001 | 552.5 | 1.25 (1.18–1.34) | <0.001 | 1.35 (1.26–1.44) | <0.001 | 1.36 (1.24–1.49) | <0.001 |
| **Number of remaining live children at loss** | | | | | | | | | | | | | |
| 0 | 228.2 | 1.19 (1.12–1.27) | <0.001 | 1.17 (1.10–1.25) | <0.001 | 1.24 (1.13–1.36) | <0.001 | 109.1 | 1.17 (1.07–1.28) | <0.001 | 1.17 (1.07–1.28) | <0.001 | 1.17 (1.02–1.34) | 0.026 |
| 1–2 | 417.4 | 1.15 (1.12–1.19) | <0.001 | 1.17 (1.14–1.21) | <0.001 | 1.17 (1.12–1.23) | <0.001 | 197.8 | 1.10 (1.05–1.15) | <0.001 | 1.14 (1.09–1.19) | <0.001 | 1.16 (1.09–1.24) | <0.001 |
| ≥3 | 722.7 | 1.37 (1.31–1.44) | <0.001 | 1.32 (1.26–1.38) | <0.001 | 1.28 (1.20–1.38) | <0.001 | 384.4 | 1.42 (1.34–1.51) | <0.001 | 1.40 (1.32–1.48) | <0.001 | 1.38 (1.26–1.50) | <0.001 |

AMI, acute myocardial infarction; CI, confidence interval; CVD, cardiovascular disease; IHD, ischemic heart disease; IRR, incidence rate ratio.

*Per 100,000 person-years.

†Model 1 was adjusted for age at follow-up.

‡Model 2 was adjusted for sex, age at follow-up, calendar year at follow-up, country of birth, educational attainment, history of psychiatric disorders and of CVDs at baseline.

‖Model 3 was adjusted for sex, age at follow-up, calendar year at follow-up, country of birth, educational attainment, marital status at baseline, income at baseline, history of psychiatric disorders and of CVDs at baseline, parents' history of CVDs, and siblings' history of CVDs.

person [8] was associated with an increased AMI risk and that loss of a child was associated with increased risks of mental illness [17], diabetes [21], atrial fibrillation [11], and death [22–24]. In contrast, other studies found no relation between the loss of a child and the risk of stroke [28], rheumatoid arthritis [29], or inflammatory bowel diseases [30]. Our findings corroborate those of Li and colleagues showing that the death of a child younger than 18 was associated with an approximately 30% increased AMI risk during the 17-year follow-up [7]. Our larger sample size, longer follow-up, and wider definition of exposure allowed us to extend the results of this earlier study by exploring, in more detail and with better precision, the importance of the child's cause of death, the child's age at loss, time since loss, and effect modification by several sociodemographic factors.

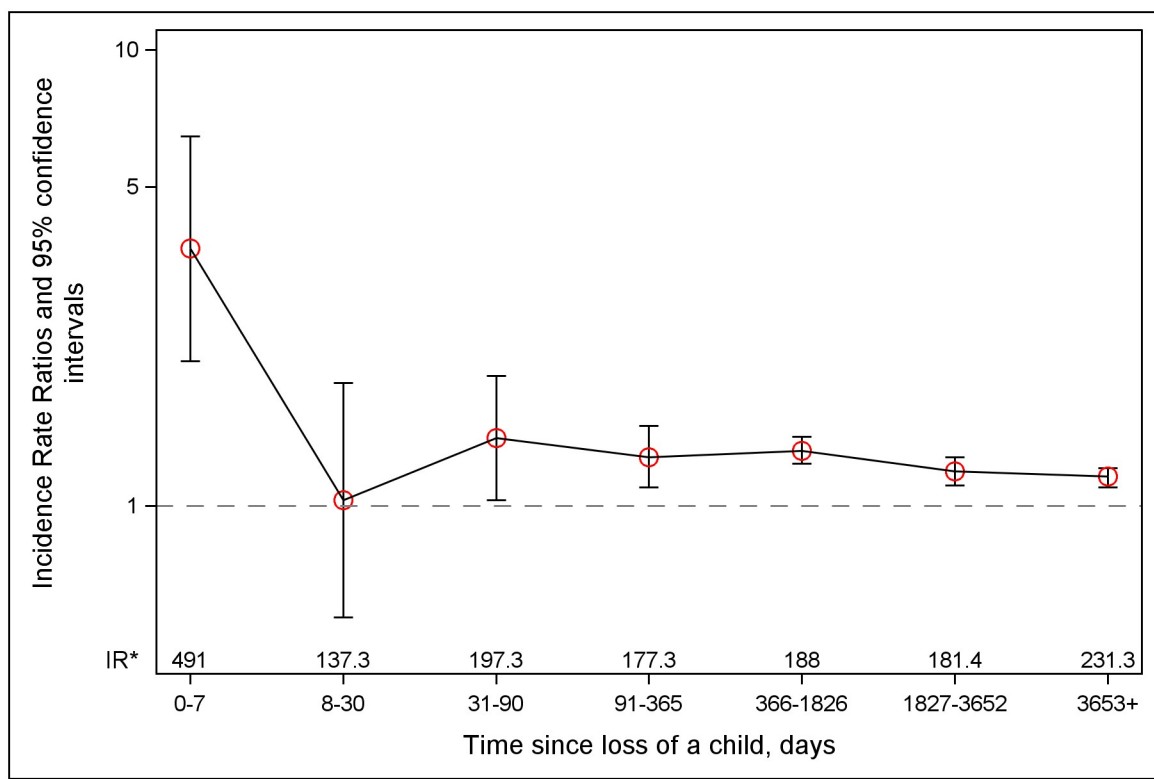

**Fig 2. IRRs and 95% CIs for AMI according to time since the death of a child.** *IR, incidence rate; the incidence rate (per 100,000 person-years) of AMI in the exposed group. AMI, acute myocardial infarction; CI, confidence interval; IRR, incidence rate ratio.

An association between the death of a child and the risk of morbidity and mortality may have at least 2 explanations. One potential explanation is that bereavement may increase the risk of morbidity and mortality through adverse stress-related lifestyle and biological changes [31]. An alternative explanation is that the genetic and environmental factors that contributed to the relative's death may increase the risk of similar diseases also in family members [31]. To try to separate the effect of grief after the death of a child from that of confounding by cardio-vascular risk factors that cluster in families, we performed analyses with exposure classified according to the child's cause of death. As expected, the associations between bereavement and IHD and AMI were strongest in case of losses of children due to CVD, suggesting that con-founding by familial cardiovascular risk factors may in part contribute to the explanation of our findings. Nevertheless, the fact that losses due to unnatural causes and of minor children, which are less likely to be prone to confounding by unmeasured cardiovascular risk factors that cluster in families, were also associated with IHD and AMI risks may suggest that stress-related mechanisms may also be of importance. Li and colleagues also reported an association between the loss of a minor child and an increased risk of AMI in the parents and that the risk was particularly high if the child died due to sudden infant death syndrome [7]. Moreover, the particularly high AMI risk during the first week after the loss of a child, which is unlikely to be explained by confounding, may also be supportive of a causal association.

The explanations for the U-shaped relationship between the child's age at loss and the parent's risk of AMI/IHD are not clear. Because parental morbidity in the prenatal period may increase the risk of several conditions that are the leading causes of infant mortality in the Western world, i.e., congenital malformations, preterm birth, birth asphyxia, and sudden

**Table 3. Adjusted IRRs and 95% CIs for the association between the death of a child and the risk of IHD in stratified analyses.**

| Subgroups | IHD | | | AMI | | |
|---|---|---|---|---|---|---|
| | Rate/per $10^5$ person-years | Multivariate IRR (95% CI) | $P$* | Rate/per $10^5$ person-years | Multivariate IRR (95% CI) | $P$* |
| **Sex**[†] | | | | | | |
| Men | 344.6 | 1.15 (1.11–1.18) | | 184.1 | 1.14 (1.09–1.18) | |
| Women | 121.2 | 1.31 (1.26–1.36) | <0.001 | 45.0 | 1.36 (1.28–1.45) | <0.001 |
| **Country** | | | | | | |
| Denmark | 249.7 | 1.12 (1.09–1.16) | | 106.9 | 1.17 (1.12–1.22) | |
| Sweden | 204.8 | 1.23 (1.18–1.27) | <0.001 | 112.6 | 1.24 (1.18–1.30) | <0.001 |
| **Age**[‡] | | | | | | |
| <50 years | 51.9 | 1.53 (1.42–1.65) | | 23.7 | 1.53 (1.36–1.71) | |
| ≥50 years | 455.4 | 1.19 (1.16–1.22) | <0.001 | 222.4 | 1.19 (1.15–1.23) | <0.001 |
| **Education**[‖] | | | | | | |
| ≤9 years | 316.1 | 1.22 (1.18–1.26) | | 159.3 | 1.21 (1.15–1.27) | |
| 10–14 | 199.2 | 1.19 (1.14–1.23) | 0.979 | 98.2 | 1.20 (1.14–1.26) | 0.801 |
| ≥15 years | 164.9 | 1.17 (1.08–1.27) | 0.666 | 71.0 | 1.20 (1.06–1.34) | 0.549 |

IRR = incidence rate ratio; CI = confidence intervals; CVD = cardiovascular disease.

* P-values are for the interaction terms between the exposure category and the effect modifier.

[†] Adjusted for age at follow-up, calendar year at follow-up, country of birth, educational attainment, history of psychiatric disorders and of cardiovascular diseases at baseline.

[‡] Adjusted for sex, age at follow-up, calendar year at follow-up, country of birth, educational attainment, history of psychiatric disorders and of cardiovascular diseases at baseline.

[‖] Adjusted for sex, age at follow-up, calendar year at follow-up, country of birth, history of psychiatric disorders and of cardiovascular diseases at baseline.

infant death syndrome [32], and because women with children born preterm [33–36], with fetal growth restriction [34] or congenital malformations [37] have increased CVD risks, we speculate that a possible explanation for the slightly stronger association for the loss of an infant than that of older children could be residual confounding by parental subclinical cardiometabolic diseases. Alternatively, as suggested also by Li and colleagues, sudden infant death syndrome, an important cause of infant mortality, may increase parents' risk of IHD also through stress-related mechanisms [7]. A sudden or unnatural death is often more stressful and more likely to be associated with complicated grief than a loss due to natural causes. Similarly, the higher IHD risk after the death of an adult child, compared with the death of a minor, not infant, child, could be due to residual confounding by parental subclinical diseases at baseline or unmeasured familial cardiovascular risk factors, or to a stronger emotional bond between parents and adult children. The latter explanation may be consistent with the observed stronger association in women than men; mothers have often stronger emotional bonds with their children and have been shown to have higher relative risks of mental illness [17], diabetes [21], and mortality [22,24] than fathers following the loss of a child. Furthermore, losing a child aged 30 years or older may also involve an increased burden for parents in terms of emotional, practical, and financial support for their affected grandchildren. Similarly, though the presence of other children at the time of the loss may help to alleviate grief [15] and buffer its adverse health effects [17,22,24], our finding that having 3 or more live children at the time of loss was associated with higher IHD and AMI risks than losing the only child—a finding similar to those reported by Li and colleagues [7] —may be indicative of higher stress arising from difficulties in combining own grief work with the care for the remaining children.

## Potential linking mechanisms

The mechanisms by which the death of a child may increase the risk of IHD and AMI may involve acute and chronic processes. Bereavement stress stimulates the activation of the hypothalamic–pituitary–adrenocortical axis and of the sympathetic nervous system, which, in turn, leads to a short-term increased inflammatory activity, elevated cortisol levels, total cholesterol, blood pressure, and heart rate, and reduction of heart rate variability and high-density lipoprotein [1,38–40], which, in turn, could trigger AMI [41]. Our results that the risk of AMI was highest in the week after the loss are in line with several studies documenting particularly high risks of mental disorders [17], AMI [5,8], stroke [5], atrial fibrillation [11,42], cardiovascular [6,10] and total mortality [22,24,31] shortly after bereavement and is supportive of a triggering effect. In addition, bereavement may induce depression, anxiety, anger, poor sleep, poor appetite, and alcohol abuse [1,40], which are associated with an increased risk of cardiac events both in the short and the long term [41]. In the long term, the death of a child is likely to result in complicated and prolonged grief [16]. Chronic bereavement stress and complicated grief may induce adverse changes in health behaviours and in endocrine, immune, vascular, and haemostatic activities contributing to further progression of atherosclerosis and thus increasing the risk of IHD and AMI. Li and colleagues found an increased AMI risk after the death of a child only from the seventh year of follow-up [7], possibly due to limited statistical power to explore short-term effects and due to the younger age of their study population. In line with the latter explanation, we also found a stronger association among the younger bereaved parents (age <50 years) than the older parents, indicating that the death of a child may contribute to the development of atherosclerosis.

## Strengths and limitations

Several strengths of our study make the findings robust, i.e., the prespecified analysis plan, the population-based design, the collection of information on exposure and outcome independently of each other, and the high quality information on mortality and the diagnoses of IHD and AMI in the 2 patient registers [43,44]. The large sample size and the long follow-up provided sufficient statistical power to perform several subanalyses that might contribute to a better understanding of the underlying mechanisms. The availability of a large number of covariates reduces residual confounding. Our study has also several limitations. First, although we adjusted for several potential confounders, we may not exclude the possibility of residual confounding from genetic factors or unmeasured socioeconomic, lifestyle, or health-related factors shared by family members. However, our analyses according to the child's cause of death showed that an increased IHD/AMI risk was present also when the child's death was due to unnatural causes, which are less likely to be affected by familial factors. Second, our findings may only apply to countries with low child mortality, well-developed free-of-charge healthcare system, and sociocultural contexts comparable to those of Denmark and Sweden.

## Conclusions

We found that the death of a child was associated with an increased risk of IHD and AMI. The findings that the association was present also in case of losses due to unnatural causes, which are less likely to be confounded by cardiovascular risk factors clustering in families, and that the risk of AMI was highest during the week after the loss suggests that stress-related mechanisms may contribute to the observed association. Our findings, if confirmed, call for intensive

surveillance and early intervention from the healthcare system among bereaved parents, particularly during the first week after the loss of a child.

## Supporting information

**S1 STROBE Checklist. STROBE Statement—Checklist of items that should be included in reports of *cohort studies*.**
(DOCX)

**S1 Analysis plan. Parental risk of AMI and IHD after the death of a child.** AMI, acute myocardial infarction; IHD, ischemic heart disease.
(DOCX)

**S1 Fig. Follow-up of study participants.**
(TIF)

**S2 Fig. The matched-control period for the hazard period of 1 month before the index event of AMI.** AMI, acute myocardial infarction.
(TIF)

**S1 Table. Population-based registers used to retrieve information for the study.**
(DOCX)

**S2 Table. The International Classification of Diseases codes used to identify the diagnoses and the causes of death.**
(DOCX)

**S3 Table. Relative risks and 95% CIs for AMI shortly after the death of a child.** AMI, acute myocardial infarction; CI, confidence interval.
(DOCX)

**S4 Table. Adjusted IRRs and 95% CIs for the association between the death of a child and the risk of IHDs in sensitivity analyses.** CI, confidence interval; IHD, ischemic heart disease; IRR, incidence rate ratio.
(DOCX)

## Author Contributions

**Conceptualization:** Imre Janszky, Krisztina D. László.

**Formal analysis:** Dang Wei.

**Funding acquisition:** Krisztina D. László.

**Investigation:** Dang Wei.

**Methodology:** Dang Wei, Imre Janszky, Fang Fang, Hua Chen, Rickard Ljung, Jiangwei Sun, Jiong Li, Krisztina D. László.

**Project administration:** Dang Wei.

**Supervision:** Imre Janszky, Fang Fang, Rickard Ljung, Jiong Li, Krisztina D. László.

**Validation:** Dang Wei, Hua Chen.

**Writing – original draft:** Dang Wei.

**Writing – review & editing:** Dang Wei, Imre Janszky, Fang Fang, Hua Chen, Rickard Ljung, Jiangwei Sun, Jiong Li, Krisztina D. László.

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
