## [Editor Report · Decision Letter 0]

19 Feb 2021

Dear Dr Wei, 

Thank you for submitting your manuscript entitled "Death of an offspring and parental risk of ischemic heart diseases: A population-based cohort study" for consideration by PLOS Medicine.

Your manuscript has now been evaluated by the PLOS Medicine editorial staff as well as by an academic editor with relevant expertise and I am writing to let you know that we would like to send your submission out for external peer review.

Kind regards,

Dr Raffaella Bosurgi

Executive Editor 

PLOS Medicine

---

## [Decision Letter · Decision Letter 1]

1 Jul 2021

Dear Dr. Wei,

Thank you very much for submitting your manuscript "Death of an offspring and parental risk of ischemic heart diseases: A population-based cohort study" (PMEDICINE-D-21-00668R1) for consideration at PLOS Medicine. 

Your paper was discussed with an academic editor with relevant expertise and sent to independent reviewers, including a statistical reviewer. The reviews are appended at the bottom of this email and any accompanying reviewer attachments can be seen via the link below:

[LINK]

In light of these reviews, we will not be able to accept the manuscript for publication in the journal in its current form, but we would like to invite you to submit a revised version that addresses the reviewers' and editors' comments fully. You will appreciate that we cannot make a decision about publication until we have seen the revised manuscript and your response, and we expect to seek re-review by one or more of the reviewers. 

We hope to receive your revised manuscript by Jul 22 2021 11:59PM. Please email us (plosmedicine@plos.org) if you have any questions or concerns.

Please let me know if you have any questions, and we look forward to receiving your revised manuscript. 

Sincerely,

Richard Turner, PhD

rturner@plos.org

To your data statement (submission form), please add web addresses for the relevant data custodians for readers interested in inquiring about access to study data.

Please combine the "Methods" and "results" subsections of your abstract. 

The final sentence of the new combined subsection should begin "Study limitations include ..." or similar and should quote 2-3 of the study's main limitations. 

In the abstract and throughout the text, please include p values alongside 95% CI, where available. 

Please remove the information on funding, competing interests and data access from between abstract and Introduction. In the event of publication, this information will appear only in the article metadata, via entries in the submission form. 

After the abstract, please add a new and accessible "Author summary" section in non-identical prose. You may find it helpful to consult one or two recent research papers published in PLOS Medicine to get a sense of the preferred style. 

Early in the Methods section of your main text, please state whether the study had a protocol or prespecified analysis plan, and if so attach the relevant document(s) as an attachment(s), referred to in the text. 

Please highlight analyses that were not prespecified. 

On p.8, please substitute "sex" for "gender" if appropriate, and at any other points in the ms.

Throughout the text, please adapt reference call-outs to the following style: "... following the loss [1,3]." (noting the absence of spaces within the square brackets). 

Please use the journal name abbreviation "PLoS ONE" in your reference list.

Please ensure that reference 29 and any others needing them have full access details. 

Please add a completed checklist for the most appropriate reporting guideline, e.g., STROBE, as an attachment, labelled "S1_STROBE_Checklist" and referred to as such in your Methods section. 

In the checklist, please refer to individual items by section (e.g., "Methods") and paragraph number, not by line or page numbers as these generally change in the event of publication. 

Comments from the reviewers:

*** Reviewer #1: 

I confine my remarks to statistical aspects of this paper. These were very well done and I have only one issue to resolve before I can recommend publication.

The authors have an entire population. This makes the use of confidence intervals a bit troublesome. CIs are about inference from a sample to a population. When you have the whole population, there is no inference to be done and the CI should all be 0 width. 

There are two options: The authors could delete the CI and not the above. On the other hand, some statisticians posit the existence of a "super-population" from which this population is randomly drawn (e.g. more countries, a longer time span, or something). I am not a big fan of this, but I wont object if the authors choose to go this route. But the issue should be addressed.

Otherwise - excellent job!

Peter Flom

*** Reviewer #2: 

The authors submitted an interesting Research Article aiming at investigating whether the death of an offspring is associated with the risk of ischemic heart disease (IHD) and acute miocardici infarction (AMI). They found a significant association in case of losses due to CVD, or other natural causes, and also in case of unnatural deaths. So they concluded that the death of an offspring was associated with an increased risk of IHD and AMI. There's findings suggests that stress-related mechanisms may also contribute to the observed associations.

The manuscript is well written, even if more attention should be paid to English grammar and structure. Several typos are present through the whole manuscript and should be addressed before any acceptance for publication. The Authors should be commended for this tremendous work but some statistical issues should be refined in order to achieve more clarity and soundness. I recommend to show all the P-Values that here are missing in the text and the tables. A comparison between baseline characteristics between groups is also advisable, and than an adjustment for baseline statistical significant differences between groups will also helpful to better understand if many confounded have been addressed or not.

*** Reviewer #3: 

This major study seeks to establish whether the death of an offspring is associated with the risk of IHD and AMI in Denmark (1973-2016) Sweden in the period 1973-2014 in the respective birth and linked health registers. 126,522 (1.9%) parents lost at least one child during the study period. The secondary outcome was 'unnatural deaths'. A total of >6.5M parents were followed. The comparison exposure group was non-bereaved parents. Parental fatal or nonfatal disease was determined in national registers. A large set of ICD codes was utilised, ranging from hypertension, diabetes to sudden coronary death. Major potential confounders of the association between parent-child bereavement are low income, low education and raised CVD risk including history of CVD. Such covariates were available in 80-95% of the population data, with gaps, e.g. little data on phenotypic CVD risk except in pregnancy. Multiple imputation could be utilized to fill in the missing data. 

There would be increased clarity in this reviewer's reading for covariate stratified results to be presented second, in table 3 rather than 2 as at present. Table 2 could then focus on attenuation of the bereavement effect by adjustment for (1) the main a priori potential confounders, then (2) a further maximally adjusted model. It is unclear why the analysis is stratified according to IHD/AMI (tables 2 and 3). There does not seem to be a clear related hypothesis, and though there some marginally significant differences, the size of effects is so similar the stratification detracts from the paper and should be confined to sensitivity analyses. If the authors wish to hypothesise a specific link between bereavement and AMI then it should be more clearly articulated and analysed, comparing the time course of effect size with other outcomes. 

The Discussion addresses the main expected topics, including key findings, past studies, bias, power and mechanisms. The precise role of the unnatural causes of death as comparator outcome could be better articulated, particularly whether this new study strengthens or weakens the accumulated evidence for a specific link between bereavement and CVD. I would prefer to see a more focused use of the terms 'bereavement stress' and 'grief' including in the abstract. Use of the general term 'stress' is widespread and can lack scientific coherence. For example there may be short or extended emotional grief responses to bereavement, followed by depressed mood, sadness and so on. Cognitive appraisal of the personal impact and meaning of the bereavement could affect health behaviours, independent of 'stress mechanisms'. 

Eric Brunner

***

[LINK]

---

## [Decision Letter · Decision Letter 2]

24 Aug 2021

Dear Dr. Wei,

Thank you very much for re-submitting your manuscript "Death of an offspring and parental risk of ischemic heart diseases: A population-based cohort study" (PMEDICINE-D-21-00668R2) for consideration at PLOS Medicine.

I have discussed the paper with our academic editor and it was also seen again by two reviewers. I am pleased to tell you that, provided the remaining editorial and production issues are fully dealt with, we expect to be able to accept the paper for publication in the journal.

[LINK]

Please let me know if you have any questions, and we look forward to receiving the revised manuscript.   

Sincerely,

Richard Turner, PhD

rturner@plos.org

Requests from Editors:

Please cite your recent paper (doi: 10.1093/eurheartj/ehaa1084) at an appropriate point in the paper (we became aware of this paper through an iThenticate check). 

Please quote summary demographic details for study participants in the abstract.

At line 16 in the abstract please make that "follow-up period".

In the Summary points, the wording of the second and final point is very similar, and we ask you to amend these points to make them less repetitive.

Under "Strengths and limitations" in the Discussion section you mention the "... prospective design" as a strength, and we ask you to remove the word "prospective". You might wish to mention your prespecified analysis plan as a strength instead.

Please substitute "sex" in place of "gender" where appropriate (e.g., in table 1).

Please spell out the institutional author name for reference 12. 

Comments from Reviewers:

*** Reviewer #1: 

The authors have addressed my concerns and I now recommend publication

Peer Flom

*** Reviewer #2: 

The manuscript significantly improved after revisions. I have no further comments or edits.

***

[LINK]

---

## [Editor Report · Decision Letter 3]

1 Sep 2021

Dear Dr Wei, 

On behalf of my colleagues and the Academic Editor, Dr Basu, I am pleased to inform you that we have agreed to publish your manuscript "Death of an offspring and parental risk of ischemic heart diseases: A population-based cohort study" (PMEDICINE-D-21-00668R3) in PLOS Medicine.

PRESS

Sincerely, 

Richard Turner, PhD 

rturner@plos.org